# Patient education and extracorporeal membrane oxygenation preferences of patients and providers in COVID care

Ethan D. Borre[1,2], Matthew L. Maciejewski[1,2,3]*, Arlene Fink[4,5], Melissa Burnside[6,7], J. Todd Purves[8], Charles D. Scales, Jr[1,7,8], Eddy Fan[9,10], Bhawandip Sandhu[10], Kevin Pignone[11], Caroline Palmer[12], Carrington Webb[11], Dana S. Guggenheim[1,13], Yuqi Zhang[3,7,14]

1 Department of Population Health Sciences, Duke University School of Medicine, Durham, North Carolina, United States of America, 2 Duke-Margolis Center for Health Policy, Duke University, Durham, North Carolina, United States of America, 3 Center for Health Services Research in Primary Care, Durham Veterans Affairs Medical Center, Durham, North Carolina, United States of America, 4 Department of Health Policy and Management, Fielding School of Public Health, University of California, Los Angeles, California, United States of America, 5 Department of Medicine, David Geffen School of Medicine, University of California, Los Angeles, California, United States of America, 6 Department of Family Medicine and Community Health, Department of Medicine, Duke University School of Medicine, Durham, North Carolina, United States of America, 7 National Clinician Scholars Program at the Clinical Research Training Program, Duke University School of Medicine, Durham, North Carolina, United States of America, 8 Division of Urology, Department of Surgery, Duke University, Durham, North Carolina, United States of America, 9 Interdepartmental Division of Critical Care Medicine and the Institute of Health Policy, Management and Evaluation, University of Toronto, Toronto, Ontario, Canada, 10 University Health Network, Toronto, Ontario, Canada, 11 University of North Carolina-Chapel Hill, Chapel Hill, North Carolina, United States of America, 12 Duke Kunshan University, Kunshan, China, 13 Duke University, Durham, North Carolina, United States of America, 14 Department of Surgery, Yale University, New Haven, Connecticut, United States of America

* mlm34@duke.edu

## Abstract

### Background

Extracorporeal membrane oxygenation (ECMO) represents an important but limited treatment for patients with severe COVID-19. We assessed the effects of an educational intervention on a person's ECMO care preference and examined whether patients and providers had similar ECMO preferences.

### Methods

In the *Video+Survey* group, patients watched an educational video about ECMO's purpose, benefits, and risks followed by an assessment of ECMO knowledge and care preferences in seven scenarios varying by hypothetical patient age, function, and comorbidities. Patients in the *Survey Only* group and providers didn't watch the video. Logistic regression was used to estimate the probability of agreement for each ECMO scenario between the two patient groups and then between all patients and providers.

**Data Availability Statement:** The authors are unable to share data publicly because of ethical and legal restrictions. First, the dataset contains potentially identifying and sensitive patient

information. Second, the ethics committees that have approved this study restrict data availability to members of the research team named in the Institutional Review Board protocol. The Duke Institutional Review Board may be contacted at (919) 668-5111 or by submitting an online form at https://irb.duhs.duke.edu/about-us/contact-us.

**Funding:** This research was funded by the National Institutes of Health (F30 DC019846) and the Duke National Clinician Scholar Program. The funders had no role in the design, conduct, collection, management, analysis, or interpretation of the data; or in the preparation, review, or approval of the manuscript. The opinions expressed are those of the authors and not necessarily those of the Department of Veterans Affairs, Duke University, the University of California - Los Angeles, the University of Toronto, the University Health Network, University of North Carolina-Chapel Hill, or Yale University.

**Competing interests:** Dr. Borre reports funding from the National Institutes of Health. Dr. Maciejewski reports research grants from the Veterans Affairs Health Services Research and Development Service (VA HSR&D, RCS 10-391), and ownership of Amgen stock due to his spouse's employment. Dr. Fan reports personal fees from Alung Technologies, Aerogen, Baxter, GE Healthcare, Inspira, and Vasomune outside the submitted work. Dr. Zhang reports funding from the Duke National Clinician Scholar Program and the Durham Veteran's Affairs. All other authors have no conflicts of interest to disclose.

## Results

*Video+Survey* patients were more likely (64% vs. 17%; p = 0.02) to correctly answer all ECMO knowledge questions than *Survey Only* patients. Patients in both groups agreed that ECMO should be considered across all hypothetical scenarios, with predicted agreement above 65%. In adjusted analyses, patients and providers had similar predicted agreement for ECMO consideration across six of the seven scenarios, but patients showed greater preference (84% vs. 41%, p = 0.003) for the scenario of a functionally dependent 65-year-old with comorbidities than providers.

## Discussion and conclusions

An educational video increased a person's ECMO knowledge but did not change their ECMO preferences. Clinicians were less likely than patients to recommend ECMO for older adults, so advanced care planning discussion between patients and providers about treatment options in critically ill patients with COVID-19 is critical.

## Introduction

SARS-CoV-2 has led to a worldwide pandemic with catastrophic effects on morbidity and mortality [1, 2]. While there is a spectrum of COVID-19 symptoms ranging from fevers to respiratory failure and cardiogenic shock, 15–30% of hospitalized patients develop acute respiratory distress syndrome (ARDS) [3, 4]. Severe cases that cannot be adequately managed with mechanical ventilation may require extracorporeal membrane oxygenation (ECMO). However, ECMO is not available in all hospitals given its resource intensity. During surging COVID-19 hospitalizations throughout 2020–2021, ECMO need has often exceeded supply in some jurisdictions [5, 6].

Studies have investigated the usage and outcomes of ECMO in COVID-19 patients [7–10], but comparatively little is known about patient knowledge and preferences around ECMO as a treatment option and how they compare to physician preferences. In prior research, patients were only slightly in favor of initiating ECMO to survive COVID-19 and usually only if they had at least a 50% chance of survival [11]. End of life (EOL) research not considering COVID-19 has demonstrated that in contrast to physicians, patients consistently choose more aggressive medical treatments at the end of life (EOL) [12–14]. The discrepancy between patients' wishes and medical decisions, and the difference between patient and physician EOL care preferences may be partially explained by low health literacy, lack of advanced directives, patient bias, and patient hope [15, 16].

Our objective was to assess the concordance in ECMO care preferences between patients and providers and the effects of an ECMO education intervention on patient knowledge and preferences. Results from this pilot study could provide initial evidence about the feasibility and potential value of an educational video intervention to increase ECMO health literacy. Concordance of patient and provider preferences for ECMO may simplify shared decision-making between caregivers and providers, while discordance in preference may indicate that greater clarity when communicating ECMO's benefits and risks may be needed to guide care of patients with COVID-19.

## Methods

### Study design

To study the effects of an ECMO educational intervention on patient knowledge and preferences, we recruited outpatient primary care patients into a prospective pilot study that randomized participants into two groups. In the *Video+Survey* group, participating patients watched a brief educational video about the purpose, benefits, and risks of ECMO followed by a survey assessment of ECMO knowledge and care preferences. In the *Survey Only* group, participating patients completed the knowledge and preference survey without watching a video or receiving any educational material on ECMO during their visit. To assess concordance of ECMO care preferences between patients and providers, we concurrently recruited Duke primary care providers as comparators. Providers were administered only the survey under the assumption that they have adequate health literacy around ECMO. At study outset, we hypothesized that a video intervention would increase concordance between patient and provider preferences for ECMO recommendation. The study was approved by the Duke Institutional Review Board.

### Educational video and survey of ECMO preferences

We adapted a publicly available patient-facing overview video of ECMO's indications, risks, and benefits video produced by the Toronto General Hospital (Toronto, Canada). The script of the final video had a Flesch-Kincaid reading grade level of 2.8. The video was uploaded on a public video domain but marked as private (see https://www.youtube.com/watch?v=lLThTPcVbcw).

We designed a three-component survey to assess ECMO knowledge and care preferences (S1 Table). The knowledge questions assessed comprehension of basic facts about ECMO presented in the video. The preference assessment presented seven hypothetical patient scenarios and asked participants for their agreement on whether ECMO should be considered. Each hypothetical ECMO scenario varied systematically by hypothetical patient age (35-year-old vs. 65-year-old), level of functional ability (independent vs. dependent), and comorbidities (present vs. absent), which are factors physicians take into account when considering ECMO for patients in clinical care [17, 18]. The survey ended with demographic questions, which were derived from United States Census questions. The final questionnaire had a Flesch-Kincaid reading grade level of 5.1. We further ensured video and survey acceptability by conducting in-depth cognitive interviews with three primary care patients prior to the initiation of the pilot trial. We subsequently incorporated their feedback regarding the comprehension and functionality of both instruments into the final products.

### Patient recruitment

Patients receiving care at a single Duke Family Medicine clinic were eligible if they were aged 50 years or older, did not have prior documented COVID-19, and did not have any significant comorbidities. We chose these exclusion criteria to understand preferences in patients who have been at greater risk of developing COVID-19 but whose preference assessments would not be influenced by personal experience with COVID. It is unknown whether personal experience with COVID would increase or decrease recommendations for ECMO use. A survivor bias may increase ECMO endorsement among those who survived an episode of severe COVID and thus we excluded these patients. Additionally, because these patients were recruited from a family medicine clinic, they may have different prior clinical experiences than those recruited from a subspecialty clinic.

Patients were screened for eligibility between July 14 and September 29, 2021, through Duke's electronic health record, and those who met inclusion/exclusion criteria were

contacted and asked for their participation. Patients who agreed to participate were randomized to either *Video+Survey* or *Survey Only* using a random number generating function (Microsoft Excel). On the day of their participation, patients were met by the research staff and were directed to a private room in the clinic. After written informed consent was obtained, participants proceeded either to the survey directly or to the video first, and then the survey. At the end of the visit, each patient was provided a coffee mug to thank them for their time. We aimed to recruit 40 patients for the pilot study given the feasibility focus of this pilot study and the challenge of recruiting patients for an in-person study during an ongoing pandemic.

### Provider recruitment

The outpatient setting is an ideal site to initiate discussions on aspects of advance care planning and family medicine providers are well situated to initiate the discussion [19]. All providers (attendings, residents, and advanced practitioners) from the Duke Family Medicine Clinic were deemed eligible for participation in the study. We chose to recruit family medicine physicians as ideally advance care planning discussion occur in this setting prior to serious illness; however, we recognize that family medicine physicians are comparatively less familiar with ECMO clinical care than critical care physicians. In September 2021, providers were sent an email that included a survey link to the consent form and the questionnaire via email. We aimed to recruit 20 providers for the pilot study given the same concerns as listed above under patient recruitment.

### Statistical analysis

We summarized demographic characteristics of the patients by randomization group and of the providers. We then reported the unadjusted proportion of patients in each group and the proportion of providers who recommended ECMO. We used logistic regression to predict the probability of recommending ECMO between the two patient groups, adjusting for baseline differences in age (50–70 years old or over 70 years old), gender, white race, education, marital status, and having a close relation (defined as a friend, spouse, partner, family member, etc.) hospitalized with COVID. A Firth correction was required due to small sample size and to correct for perfect prediction of several explanatory variables [20]. We computed marginal effects of the predicted probability of recommending ECMO in each patient group for the seven scenarios, with all other covariates set to their means, because odds ratios are not directly interpretable on the probability scale [21]. We reported p-values for the computed odds ratio of the logistic regression for each comparison.

In separate logistic regressions with a Firth correction, the probability of recommending ECMO between providers and the pooled patient groups was estimated, adjusting for gender, white race, marital status, and having a close relation hospitalized with COVID. Gender and education were not included as covariates because of collinearity with the main effect of being a provider or patient respondent. Marginal effects were also generated in this comparison using methods similar to those described above. All analyses were completed in STATA version 17.0.

## Results

During the recruitment period, 231 patients met inclusion criteria and were contacted for recruitment, of which 108 (47%) patients did not respond and 56 (24%) patients declined to participate. The remaining 67 (29%) patients agreed to participate in the study but 26 of them did not complete the pilot study due to scheduling and logistical reasons. Of the 41 (18%) patients who completed the study, 23 patients were randomized to *Video+Survey* and 18

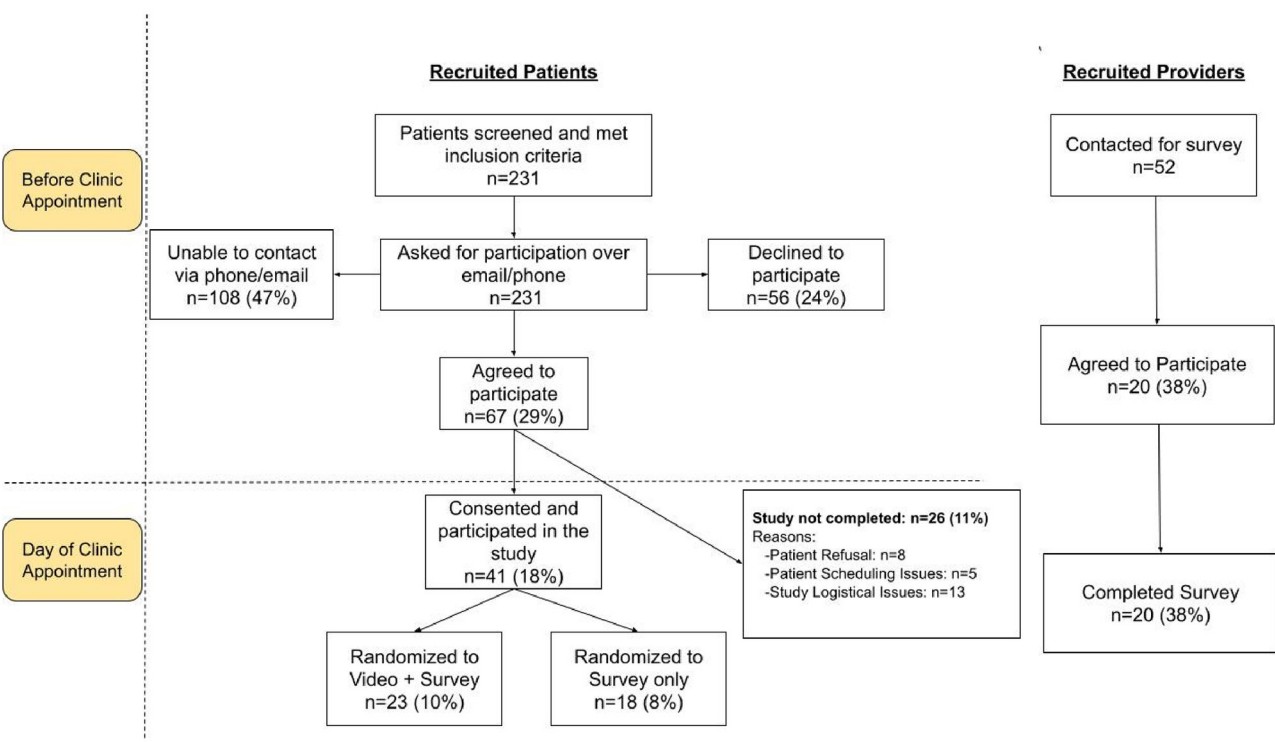

**Fig 1. Shows the flow diagram for patient and provider recruitment to the study.**

patients were randomized to *Survey Only*. We emailed 52 total providers, of whom 20 (38%) agreed to participate and completed the emailed survey (Fig 1). Patient group and provider demographics are displayed in Table 1. Notably, 17% of patients in the *Survey Only* group and 39% of patients in the *Video+Survey* group reported having a close friend or family member hospitalized for COVID-19 in the recent past.

## Patient ECMO knowledge and care preferences

Patients randomized to the *Video+Survey* group were more likely (64% vs. 17%, Table 1) to correctly answer all four ECMO knowledge questions, which also held in adjusted analyses (p = 0.02, Table 2).

Patients in both groups generally agreed that ECMO should be considered across all hypothetical ECMO scenarios, with unadjusted rates of agreement not falling below 65%. For the 35-year-old scenarios, 72–83% of patients in the *Survey Only* group agreed that ECMO should be considered, with agreement varying trivially based on comorbidity and dependency status, whereas patients who were randomized to *Video+Survey* agreed 91–96% of the time. For the 65-year-old scenarios, agreement ranged between 72–89% for the *Survey Only* group, and 91–100% for the *Video+Survey* group. Provider agreement was 70–90% for the 35-year-old scenarios, and 45–85% for the 65-year-old scenarios. In adjusted analyses of all seven hypothetical scenarios, patients who saw the ECMO education video before completing the survey had statistically similar rates in agreeing ECMO be considered as patients who did not see the video (Table 2).

**Table 1. Demographic characteristics of participating patients and providers.**

| | Patients (n = 41) Randomized to: | | Providers (n = 20) |
|---|---|---|---|
| | **Survey Only (n = 18, 43.9%)** | **Video +Survey (n = 23, 56.1%)** | |
| **Age** | | | |
| <50 years | - | - | 13 |
| 50–69 years | 8 (44%) | 14 (61%) | 4 |
| >=70 years | 10 (56%) | 9 (39%) | 1 (5%) |
| Prefer not to say | 0 | 0 | 3 (15%) |
| **Gender** | | | |
| Man | 6 (33%) | 9 (39%) | 5 (25%) |
| Woman | 12 (67%) | 14 (61%) | 14 (70%) |
| Other | 0 | 0 | 1 (5%) |
| **Race/Ethnicity** | | | |
| White | 15 (83%) | 17 (74%) | 12 (60%) |
| Other | 3 (17%) | 6 (26%) | 8 (40%) |
| **Education** | | | |
| Bachelor's Degree or less | 11 (61%) | 11 (48%) | 0 |
| Master's Degree | 3 (17%) | 7 (30%) | 2 (10%) |
| Doctorate Degree (Ex: Ph.D., MD, J.D) | 3 (17%) | 2 (9%) | 18 (90%) |
| Prefer not to say | 1 (6%) | 1 (4%) | 0 |
| **Married** | | | |
| Currently married | 11 (61%) | 14 (61%) | 13 (65%) |
| Currently not married | 7 (39%) | 9 (39%) | 7 (35%) |
| **Close friend or relative hospitalized with COVID** | 3 (17%) | 9 (39%) | 3 (15%) |
| **Correctly answer all ECMO knowledge questions** | 3 (17%) | 15 (65%) | N/A |

**Table 2. Predicted probability of participating patients agreeing extracorporeal membrane oxygenation (ECMO) should be considered in the hypothetical scenario.**

| | Survey Only (n = 18) | Video +Survey (n = 23) | p-value |
|---|---|---|---|
| Answering all 4 ECMO knowledge questions correctly | 17% | 64% | 0.02 |
| 35-year-old female, lives independently | 77% | 96% | 0.13 |
| 35-year-old female, lives independently, takes 3 pills/day for diabetes and high BP | 82% | 95% | 0.32 |
| 35-year-old female, because of weakness in arms and legs, she needs an aide to help her bathe and dress | 74% | 92% | 0.17 |
| 35-year-old female, because of weakness in arms and legs, she needs an aide to help her bathe and dress, takes 3 pills/day for diabetes and high BP | 75% | 97% | 0.16 |
| 65-year-old female, lives independently | 78% | 96% | 0.30 |
| 65-year-old female, lives independently, takes 3 pills/day for diabetes and high BP | 74% | 94% | 0.21 |
| 65-year-old female, because of weakness in arms and legs, she needs an aide to help her bathe and dress, takes 3 pills/day for diabetes and high BP | 65% | 94% | 0.14 |

Note: Logistic regression adjusted for age, gender, race, education, and marital status

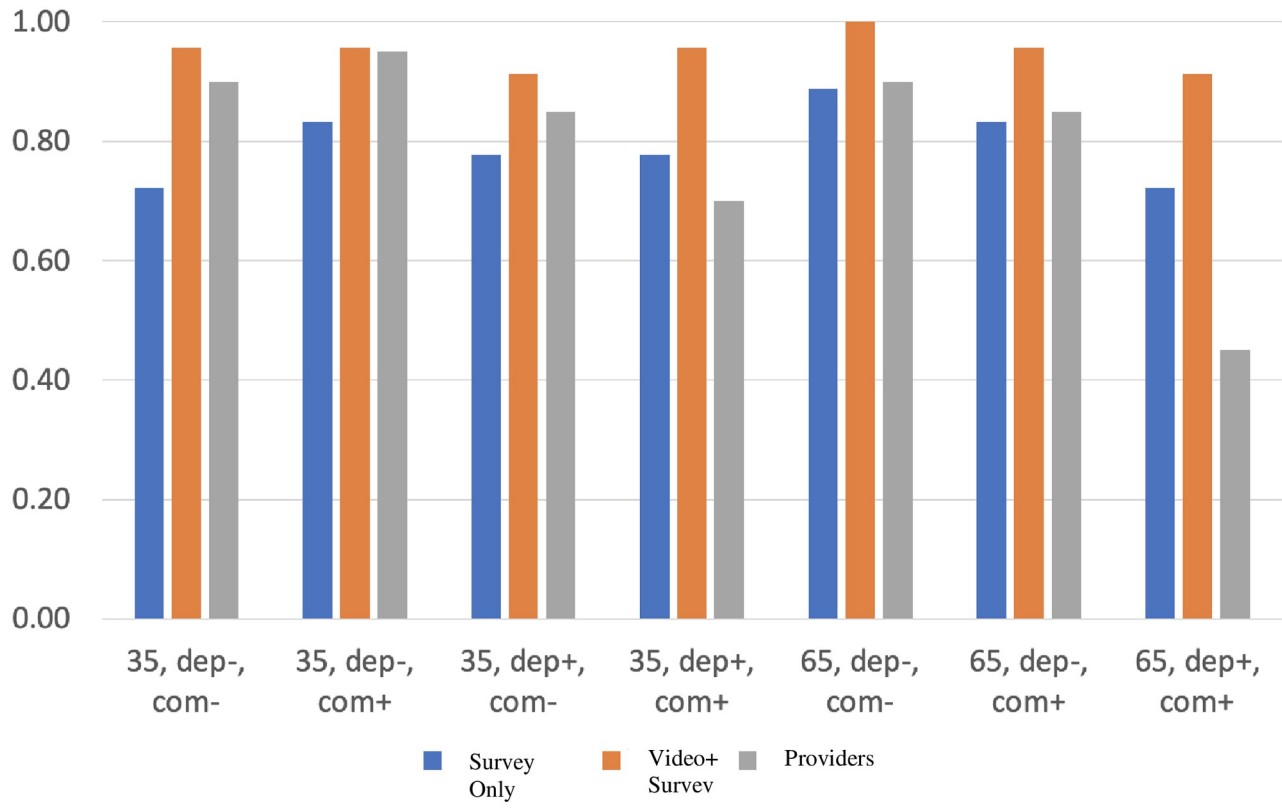

| 35-year-old female, lives independently |
| 35-year-old female, lives independently, takes 3 pills a day for diabetes and high blood pressure |
| 35-year-old female, because of weakness in her arms and legs, she needs an aide to help her bathe/dress |
| 35-year-old female, because of weakness in her arms and legs, she needs an aide to help her bathe/dress, takes 3 pills a day for diabetes and high blood pressure |
| 65-year-old female, lives independently |
| 65-year-old female, lives independently, takes 3 pills a day for diabetes and high blood pressure |
| 65-year-old female, because of weakness in her arms and legs, she needs an aide to help her bathe/dress, takes 3 pills a day for diabetes and high blood pressure |

**Fig 2. Depicts the proportions of participating patients and providers (on the y-axis) who endorse extracorporeal membrane oxygenation (ECMO) for differing scenarios (presented on the x-axis).** The different patient groups are represented by the colored bars: blue for the *Survey Only* group, orange for *Video+Survey*, and grey for providers. In the scenarios, dep is short for dependence, and com is short for comorbidity. The scenarios list the age, whether the described person is dependent+/-, and has comorbidities +/-. The legend below the figure includes the full descriptions of the patient scenarios.

### Patient and provider ECMO care knowledge and preferences

In adjusted analyses that pooled patients in the two groups, providers were more likely (Fig 2; 58% vs 43%, p = 0.28) to answer all four ECMO knowledge questions correctly than patients but this difference was not statistically significant. Providers and patients had similar predicted agreement for ECMO consideration across six of the seven hypothetical ECMO scenarios. For the 65-year-old with functional dependency and comorbidities scenario, patients had predicted probabilities of recommending ECMO that were more than twice as high (84% vs. 41%, p = 0.003) as that of providers (Table 3).

**Table 3. Predicted probability of participating patients and providers agreeing extracorporeal membrane oxygenation (ECMO) should be considered in the hypothetical scenario.**

| | All Patients (n = 41) | Providers (n = 20) | p-value |
|---|---|---|---|
| Answering all 4 ECMO knowledge questions correctly | 43% | 58% | 0.28 |
| 35-year-old female, lives independently | 84% | 88% | 0.62 |
| 35-year-old female, lives independently, takes 3 pills/day for diabetes and high BP | 93% | 92% | 0.91 |
| 35-year-old female, because of weakness in arms and legs, she needs an aide to help her bathe and dress | 85% | 84% | 0.95 |
| 35-year-old female, because of weakness in arms and legs, she needs an aide to help her bathe and dress, takes 3 pills/day for diabetes and high BP | 87% | 71% | 0.14 |
| 65-year-old female, lives independently | 94% | 90% | 0.53 |
| 65-year-old female, lives independently, takes 3 pills/day for diabetes and high BP | 89% | 84% | 0.56 |
| 65-year-old female, because of weakness in arms and legs, she needs an aide to help her bathe and dress, takes 3 pills/day for diabetes and high BP | 84% | 41% | 0.003 |

Note: Logistic regression adjusted for gender, race, and marital status

## Discussion

ECMO is an invasive and resource-intensive intervention for severe ARDS that has become a critical resource during the pandemic. The need for ECMO vastly exceeds supply in the hospitals that have it and is not available in all hospitals [6]. ECMO may have considerable benefits for those not doing well with mechanical ventilation, but it is not without significant risks as well [8, 22]. The supply constraints of ECMO have required doctors to make difficult clinical decisions in the absence of clinical guidelines, which have resulted in disagreements between caregivers and providers regarding appropriateness for a given patient [5].

This study was motivated by the lack of evidence about the degree to which patients/caregivers and providers agree on ECMO prioritization for patients who vary in risk factors and builds upon a prior study that obtained public perceptions about ECMO but did not examine patient-provider concordance in ECMO preferences [11]. We found that patients recommended ECMO at high rates (65–82%), particularly if they saw an educational video before sharing their ECMO care preferences (92–97%). Seeing a video resulted in meaningful differences (13–29% higher) in care preferences between the two patient groups, but they were not statistically significant in this small sample (n = 41). In the *Video+Survey* group, ECMO care preferences were constant despite increases in the age, functional dependence, and comorbidity burden of hypothetical patients. We hypothesize several reasons why patients in the *Video+Survey* subgroup endorsed ECMO at higher rates, including that the risks were not as prominently described as benefits in the video, the video emphasized the positive aspects of ECMO, patient attachment to hope, and that patients could have perceived themselves to be similar to the hypothetical older patients. It is also possible that the high rates of patient endorsement of ECMO across all seven hypothetical scenarios may be due to patient's difficulty with choosing "life and death" interventions (like ECMO), irrespective of knowledge, due to the gravity or immediacy of not recommending them.

We also found that enrolled patients had similar care preferences for six of the seven hypothetical ECMO scenarios as providers, but that patients were twice as likely (84% vs 41%) to recommend ECMO consideration for a 65-year-old with functional dependency and comorbidities. This discordance may be due to providers having a more accurate estimate of ECMO's success in this population, based on a better understanding of benefits and especially

of risks, as well as effects of patient hope [16]. These results support reports of discordance between patients and providers, which suggests that enhanced communication between providers and patients might allow for clearer explanations of risks and benefits to facilitate patient decision making that prioritizes their values [5].

Lastly, we found that patients who viewed the ECMO education video displayed greater knowledge than those who did not view the video, even though it did not change their preferences for ECMO. This suggests a short educational video may increase patient and caregiver awareness of ECMO's risks and benefits, which may facilitate shared decision-making between patients and providers. Patients who received the video intervention appeared more likely to recommend ECMO consideration compared to those who did not receive the intervention (though not statistically significant in this small sample), which merits exploration in future work to understand these differences. Future research should clarify the possibility that education around the risks and benefits of ECMO increases comfort with the procedure and potentially increase the likelihood of patients, caregivers, and providers opting for this treatment. Reducing discordance in patient and provider preferences in this way can alleviate conflict especially in a pandemic that requires difficult choices about resource allocation.

There are an important number of limitations to consider in this pilot study. First, we demonstrated it was feasible to provide an educational video and survey to patients seen in primary care and limited to age 50 and over and to primary care providers, but younger patients, hospitalists, or critical care specialists may have very different care preferences than those reported here. Future work should explore similar interventions in these other populations. Second, patients in the two groups had significant differences (approximately double) in a history of a family member or friend who was hospitalized for COVID-19, which might bias the estimated differences in ECMO knowledge, which we attempted to adjust for. That said, we did not inquire about any prior personal or close relation experience with intubation, ECMO, or recent death which may have also differed between the groups. Third, a larger sample over multiple clinics across different medical specialties would provide greater statistical power and enable subgroup analyses, which were infeasible in this pilot study. There was likely non-random recruitment into the group of patients that agreed to participate in the study, mostly due to lack of patient response to the recruitment phone call. This may have biased the final patient sample to those with no history of COVID who were more trusting of the healthcare system and future studies may consider a different recruitment process. Fourth, the patient assessment of ECMO knowledge was fairly simplistic and could be made more comprehensive in future work. Finally, it is possible that the patient preferences were greatly influenced by the presentation of benefits and risks, and care preferences may have been different if the video emphasized risks over benefits.

## Conclusion

In conclusion, this pilot study found that an educational video can increase patient knowledge of the benefits and risks of ECMO. There are meaningful differences in ECMO care preferences between patients and providers for high-risk patients that may be addressable by targeted patient education.

## Supporting information

**S1 Table. Survey about participating patient and provider knowledge, attitudes, and preferences about Extra Corporeal Membrane Oxygenation (ECMO).**
(DOCX)

## Author Contributions

**Conceptualization:** Matthew L. Maciejewski, J. Todd Purves, Yuqi Zhang.

**Data curation:** Ethan D. Borre, Matthew L. Maciejewski, Arlene Fink, Yuqi Zhang.

**Formal analysis:** Ethan D. Borre, Matthew L. Maciejewski.

**Funding acquisition:** Matthew L. Maciejewski.

**Investigation:** Ethan D. Borre, Matthew L. Maciejewski, Arlene Fink, Melissa Burnside, Bhawandip Sandhu, Yuqi Zhang.

**Methodology:** Ethan D. Borre, Matthew L. Maciejewski, Arlene Fink, Eddy Fan, Yuqi Zhang.

**Project administration:** Ethan D. Borre, Matthew L. Maciejewski, Bhawandip Sandhu, Kevin Pignone, Caroline Palmer, Carrington Webb, Dana S. Guggenheim, Yuqi Zhang.

**Resources:** Bhawandip Sandhu.

**Supervision:** Ethan D. Borre, Matthew L. Maciejewski, Arlene Fink, Melissa Burnside, J. Todd Purves, Charles D. Scales, Jr, Eddy Fan, Yuqi Zhang.

**Validation:** Matthew L. Maciejewski.

**Writing – original draft:** Ethan D. Borre, Matthew L. Maciejewski, Yuqi Zhang.

**Writing – review & editing:** Ethan D. Borre, Matthew L. Maciejewski, Arlene Fink, Melissa Burnside, J. Todd Purves, Charles D. Scales, Jr, Eddy Fan, Bhawandip Sandhu, Kevin Pignone, Caroline Palmer, Carrington Webb, Dana S. Guggenheim, Yuqi Zhang.

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
