## [Decision Letter · Decision Letter 0]

13 Mar 2023

PONE-D-23-01032Patient Education and Extracorporeal Membrane Oxygenation Preferences of Patients and Providers in COVID CarePLOS ONE

Dear Dr. Maciejewski,

Thank you for submitting your manuscript to PLOS ONE. After careful consideration, we feel that it has merit but does not fully meet PLOS ONE’s publication criteria as it currently stands. Therefore, we invite you to submit a revised version of the manuscript that addresses the points raised during the review process.

We look forward to receiving your revised manuscript.

Kind regards,

Omar A. Almohammed, Ph.D.

Academic Editor

PLOS ONE

Journal Requirements:

 “This research was funded by the National Institutes of Health (F30 DC019846) and the Duke National Clinician Scholar Program.”

“Mr. Borre reports funding from the National Institutes of Health. Dr. Maciejewski reports research grants from the Veterans Affairs Health Services Research and Development Service (VA HSR&D, RCS 10-391), and ownership of Amgen stock due to his spouse’s employment. Dr. Fan reports personal fees from Alung Technologies, Aerogen, Baxter, GE Healthcare, Inspira, and Vasomune outside the submitted work. Dr. Zhang reports funding from the Duke National Clinician Scholar Program and the Durham Veteran’s Affairs. All other authors have no conflicts of interest to disclose.”

“Mr. Borre reports funding from the National Institutes of Health. Dr. Maciejewski reports research grants from the Veterans Affairs Health Services Research and Development Service (VA HSR&D, RCS 10-391), and ownership of Amgen stock due to his spouse’s employment. Dr. Fan reports personal fees from Alung Technologies, Aerogen, Baxter, GE Healthcare, Inspira, and Vasomune outside the submitted work. Dr. Zhang reports funding from the Duke National Clinician Scholar Program and the Durham Veteran’s Affairs.”

“This research was funded by the National Institutes of Health (F30 DC019846) and the Duke National Clinician Scholar Program.”

7. In your Data Availability statement, you have not specified where the minimal data set underlying the results described in your manuscript can be found. PLOS defines a study's minimal data set as the underlying data used to reach the conclusions drawn in the manuscript and any additional data required to replicate the reported study findings in their entirety. All PLOS journals require that the minimal data set be made fully available. For more information about our data policy, please see http://journals.plos.org/plosone/s/data-availability.

8. Your ethics statement should only appear in the Methods section of your manuscript. If your ethics statement is written in any section besides the Methods, please delete it from any other section.

9. Please ensure that you refer to Figure 2 in your text as, if accepted, production will need this reference to link the reader to the figure.

Reviewers' comments:

Reviewer's Responses to Questions

**Comments to the Author**

1. Is the manuscript technically sound, and do the data support the conclusions?

Reviewer #1: No

Reviewer #2: Yes

2. Has the statistical analysis been performed appropriately and rigorously? 

Reviewer #1: No

Reviewer #2: Yes

3. Have the authors made all data underlying the findings in their manuscript fully available?

Reviewer #1: No

Reviewer #2: No

4. Is the manuscript presented in an intelligible fashion and written in standard English?

Reviewer #1: Yes

Reviewer #2: Yes

5. Review Comments to the Author

Reviewer #1: ECMO is an advanced form of life support that requires rigorous training, especially in operation training. Since ECMO is not widely used in clinical practice, the teaching of ECMO needs to rely on some video means. In the period of COVID-19, the number of patients eligible for ECMO was large, but doctors with enough experience are limited. Training should be to use these video resources to conduct training more efficiently, instead of not relying on them. Undoubtedly, the consequences of not using these measures can be imagined. This study has limited clinical value.

Reviewer #2: Manuscript: PONE-D-23-01032

Title: Patient Education and Extracorporeal Membrane Oxygenation Preferences of Patients

and Providers in COVID Care

General Comments: ECMO is a vital but limited resource that has significant use and impact for severely ill COVID patients with failed oxygenation. At the peak of the pandemic, and during surges, ECMO availability became strained and limited. As such hard decisions needed to be made as to best patients to place on support - balancing patient desire, provider recommendation and equity. The authors conducted a study comparing patients to providers preferences for ECMO via use of a survey and video education tool Three groups were established: Patient - with education video + survey; Patients with Survey only; Providers with survey. A Range of scenarios from younger age to increasing age were employed associated with survey questions. Comparison of outcomes (consideration/preference for ECMO) were compared across groups as well as between patients with and w/o video, and physicians to patients. Patients educated via video had a higher rate of answering questions about the procedure with greater accuracy, with both groups having similar agreement as to ECMO recommendations across 6/7 scenarios. Results between patients were similar to providers. One key distinction emerged for hypothetical older adult patients - there physicians were less likely to recommend ECMO than patients.

This study provides useful information as to the impact of training video and education and discussion in general with patients regarding selection of high risk and scarce resources. Despite this there are several issues that should be addressed to strengthen this paper.

General issues to be discussed:

1. This study was performed on well patients - purposely avoiding those with a COVID history. Authors should comment on why this group was selected. Also they should comment on expected mindset and impact of those with a COVID Hx vs. those w/o as to how patients would likely respond, and their biases. This would be useful for the reader.

2. The authors should define in methods and results what prior education was done with the patient only group. Did they learn from providers, did they learn from other sources. What was their knowledge and impression of ECMO prior to coming into the study?

3. The study would benefit in the manuscript from a Hypothesis

Specific points to address:

**Introduction:**

Pg 3. “The discrepancy between patients’ wishes and medical decisions, and the difference between patient and physician EOL care preferences may be partially explained by low health literacy and lack of advanced directives.

This difference may also relate to patient bias and patient hope. This should be added here as well as references for these points. This should also be discussed in the discussion.

**Methods:**

Pg 4. Recruitment of primary care patients

Beyond the fact of a readily available group please discuss any issues or bias that this group would have versus those in family medicine, internal medicine or for that matter subspecialty clinics. Please comment here and in discussion.

Pg 4. Engagement of Primary Care providers

Similar point as to the patients. This is certainly a valid and useful cohort. Please comment on and discuss bias intrinsic in this group as they are NOT domain experts re ECMO , i.e.. Compared to ICU/CCU/Intensivists/Cardiologists or CT surgeons. Contrast to these groups, even though theoretic, should be addressed.

Pg 5. Patient recruitment

“We chose these exclusion criteria to understand preferences in patients who have been at greater risk of developing COVID-19 but whose preference assessments would not be influenced by personal experience with COVID.”

Would be useful to know if they had family members or close friends with severe COVID? Anyone intubated? Anyone on ECMO? Anyone close died? As all of this will inject subliminal bias. Great if data could be provided. Regardless, this should also be discussed in the discussion.

**Results:**

Pg 7. “During the recruitment period, 231 patients met inclusion criteria, of which 108

Would be useful to know how many total patients were approached/interviewed

This is important as there is progressive narrowing that has occurred in this study with progressive deselection resulting in a small group. This progressive whittling effect impacts the mindset of the final participants and should be discussed.

Pgs 7 and 8. “Notably, 17% of patients in the Survey Only group and 39% of patients in the Video+Survey group reported having a close friend or family member hospitalized for COVID-19 in the recent past.”

Please discuss the influence of a 2X increased connection in this group to close members/friends with COVID as to bias and outcome. Add to the Discussion section.

**Discussion:**

Pg 10. “patients were given little time to reflect on the benefits and harms before completing the preference portion of the survey, and patients could have perceived themselves to be similar to the hypothetical older patients.”

Additional reasons here include: 1. the video group getting a sense of the positive benefit of ECMO and 2. an attachment to hope. These points should be added and discussed.

Pg 10. “These results support reports of discordance between patients and providers, and efforts should be made to communicate risks to patients and caregivers to minimize discordance.”

This sentence should be modified and softened. It conveys to the reader a bit of intimidation. It conveys that an attempt should be made to make patients agree with providers. Patients after all have choice, and desire hope. Care should be exerted in the tone throughout the paper, otherwise it comes off as one-sided.

6. PLOS authors have the option to publish the peer review history of their article (what does this mean?). If published, this will include your full peer review and any attached files.

Reviewer #1: No

Reviewer #2: Yes

---

## [Author Response · Author response to Decision Letter 0]

8 May 2023

May 1, 2023

Omar A. Almohammed, PhD

Academic Editor

PLOS ONE

Dear Dr. Almohammed,

We are pleased to submit a revised version of our manuscript, “Patient Education and Extracorporeal Membrane Oxygenation Preferences of Patients and Providers in COVID Care” [PONE-D-23-01032] for consideration at PLOS One.

We are grateful to the two reviewers of the manuscript for their helpful comments, and we have made revisions that address all of the reviewers’ comments that we believe significantly improve this manuscript. Detailed responses to those comments are provided in the attached letter. Throughout we include examples of the changed text in the revised manuscript in bold.

Thank you for your consideration of the revised manuscript.

Sincerely,

Matthew L. Maciejewski, PhD

Professor

Department of Population Health Sciences

Duke University Medical Center

~~~~~~~~~~~~

Reviewer #1

Overall Comments: ECMO is an advanced form of life support that requires rigorous training, especially in operation training. Since ECMO is not widely used in clinical practice, the teaching of ECMO needs to rely on some video means. In the period of COVID-19, the number of patients eligible for ECMO was large, but doctors with enough experience are limited. Training should be to use these video resources to conduct training more efficiently, instead of not relying on them. Undoubtedly, the consequences of not using these measures can be imagined. This study has limited clinical value. 

Response: We thank you for your review of our manuscript and agree that the number of patients eligible for ECMO vastly exceeded doctors with sufficient experience to optimally allocate this scarce treatment. That realization in the middle of the pandemic was one of the inspirations for conducting this study.

Reviewer #2

Overall Comments: ECMO is a vital but limited resource that has significant use and impact for severely ill COVID patients with failed oxygenation. At the peak of the pandemic, and during surges, ECMO availability became strained and limited. As such hard decisions needed to be made as to best patients to place on support - balancing patient desire, provider recommendation and equity. The authors conducted a study comparing patients to providers preferences for ECMO via use of a survey and video education tool Three groups were established: Patient - with education video + survey; Patients with Survey only; Providers with survey. A Range of scenarios from younger age to increasing age were employed associated with survey questions. Comparison of outcomes (consideration/preference for ECMO) were compared across groups as well as between patients with and w/o video, and physicians to patients. Patients educated via video had a higher rate of answering questions about the procedure with greater accuracy, with both groups having similar agreement as to ECMO recommendations across 6/7 scenarios. Results between patients were similar to providers. One key distinction emerged for hypothetical older adult patients - there physicians were less likely to recommend ECMO than patients.

This study provides useful information as to the impact of training video and education and discussion in general with patients regarding selection of high risk and scarce resources. Despite this there are several issues that should be addressed to strengthen this paper.

Response: Thank you for reviewing our manuscript and providing the constructive critiques below that improved clarity of the presentation considerably.

Specific Comments:

1. This study was performed on well patients - purposely avoiding those with a COVID history. Authors should comment on why this group was selected. Also they should comment on expected mindset and impact of those with a COVID Hx vs. those w/o as to how patients would likely respond, and their biases. This would be useful for the reader.

Response: We now comment further on potential biases that we avoided by excluding persons with a history of prior COVID infection in the Methods:

• In the Patient Recruitment section of methods (page 6), we clarify that we chose these exclusion criteria to understand preferences in patients who have been at greater risk of developing COVID-19 but whose preference assessments would not be influenced by personal experience with COVID. It is unknown whether personal experience with COVID would increase or decrease recommendations for ECMO use. A survivor bias may increase ECMO endorsement among those who survived an episode of severe COVID and thus we excluded these patients. We also noted in this sample restriction in the limitation section of the discussion. 

2. The authors should define in methods and results what prior education was done with the patient only group. Did they learn from providers, did they learn from other sources. What was their knowledge and impression of ECMO prior to coming into the study?

Response: We now clarify on page 4 of the Methods section that the Survey Only group did not receive any other education during this visit in the Methods:

• In the Survey Only group, participating patients completed the knowledge and preference survey without watching a video or receiving any educational material on ECMO during their visit.

3. The study would benefit in the manuscript from a Hypothesis

Response: We have now added our original hypothesis in the Methods on page 4:

• At study outset, we hypothesized that a video intervention would increase concordance between patient and provider preferences for ECMO recommendation.

Introduction:

4. Pg 3. “The discrepancy between patients’ wishes and medical decisions, and the difference between patient and physician EOL care preferences may be partially explained by low health literacy and lack of advanced directives.” This difference may also relate to patient bias and patient hope. This should be added here as well as references for these points. This should also be discussed in the discussion.

Response: We have added these points on page 3 of the Introduction:

• The discrepancy between patients’ wishes and medical decisions, and the difference between patient and physician EOL care preferences may be partially explained by low health literacy, lack of advanced directives, patient bias, and patient hope.15,16

And on page 11 of the Discussion:

• This discordance may be due to providers having a more accurate estimate of ECMO’s success in this population, based on a better understanding of benefits and especially of risks, as well as effects of patient hope.

Methods:

5. Pg 4. Recruitment of primary care patients

Beyond the fact of a readily available group please discuss any issues or bias that this group would have versus those in family medicine, internal medicine or for that matter subspecialty clinics. Please comment here and in discussion.

Response: We have now added this consideration into the Methods on pages 6-7:

• Additionally, because these patients were recruited from a family medicine clinic, they may have different prior clinical experiences than those recruited from a subspecialty clinic.

And on page 12 of Discussion:

• Third, a larger sample over multiple clinics across different medical specialties would provide greater statistical power and enable subgroup analyses, which were infeasible in this pilot study.

6. Pg 4. Engagement of Primary Care providers

Similar point as to the patients. This is certainly a valid and useful cohort. Please comment on and discuss bias intrinsic in this group as they are NOT domain experts re ECMO , i.e.. Compared to ICU/CCU/Intensivists/Cardiologists or CT surgeons. Contrast to these groups, even though theoretic, should be addressed.

Response: We now comment on this difference on pages 6-7 of the Methods:

• We chose to recruit family medicine physicians as ideally advance care planning discussion occur in this setting prior to serious illness; however, we recognize that family medicine physicians are comparatively less familiar with ECMO clinical care than critical care physicians.

7. Pg 5. Patient recruitment

“We chose these exclusion criteria to understand preferences in patients who have been at greater risk of developing COVID-19 but whose preference assessments would not be influenced by personal experience with COVID.” Would be useful to know if they had family members or close friends with severe COVID? Anyone intubated? Anyone on ECMO? Anyone close died? As all of this will inject subliminal bias. Great if data could be provided. Regardless, this should also be discussed in the discussion.

Response: We have included this point on page 12 of the Discussion:

• That said, we did not inquire about any prior personal or close relation experience with intubation, ECMO, or recent death which may have also differed between the groups.

Results:

8. Pg 7. “During the recruitment period, 231 patients met inclusion criteria, of which 108

Would be useful to know how many total patients were approached/interviewed. This is important as there is progressive narrowing that has occurred in this study with progressive deselection resulting in a small group. This progressive whittling effect impacts the mindset of the final participants and should be discussed.

Response: We have now added clarified the recruitment process on page 8 of the Results:

• During the recruitment period, 231 patients met inclusion criteria and were contacted for recruitment, of which 108 (47%) patients did not respond and 56 (24%) patients declined to participate. The remaining 67 (29%) patients agreed to participate in the study but 26 of them did not complete the pilot study due to scheduling and logistical reasons.

And page 12 of Discussion:

• There was likely non-random recruitment into the group of patients that agreed to participate in the study, mostly due to lack of patient response to the recruitment phone call. This may have biased the final patient sample to those with no history of COVID who were more trusting of the healthcare system and future studies may consider a different recruitment process.

9. Pgs 7 and 8. “Notably, 17% of patients in the Survey Only group and 39% of patients in the Video+Survey group reported having a close friend or family member hospitalized for COVID-19 in the recent past.” Please discuss the influence of a 2X increased connection in this group to close members/friends with COVID as to bias and outcome. Add to the Discussion section.

We have now included this on page 12 of the Discussion:

• Second, patients in the two groups had significant differences (approximately double) in a history of a family member or friend who was hospitalized for COVID-19, which might bias the estimated differences in ECMO knowledge, which we attempted to adjust for.

Discussion:

10. Pg 10. “patients were given little time to reflect on the benefits and harms before completing the preference portion of the survey, and patients could have perceived themselves to be similar to the hypothetical older patients.” Additional reasons here include: 1. the video group getting a sense of the positive benefit of ECMO and 2. an attachment to hope. These points should be added and discussed.

Response: We have now included these possibilities on page 10 of the Discussion:

• We hypothesize several reasons why patients in the Video+Survey subgroup endorsed ECMO at higher rates, including that the risks were not as prominently described as benefits in the video, the video emphasized the positive aspects of ECMO, patient attachment to hope, and that patients could have perceived themselves to be similar to the hypothetical older patients.

11. Pg 10. “These results support reports of discordance between patients and providers, and efforts should be made to communicate risks to patients and caregivers to minimize discordance.” This sentence should be modified and softened. It conveys to the reader a bit of intimidation. It conveys that an attempt should be made to make patients agree with providers. Patients after all have choice, and desire hope. Care should be exerted in the tone throughout the paper, otherwise it comes off as one-sided.

Response: We have now amended this sentence as suggested on page 11 of the Discussion:

• These results support reports of discordance between patients and providers, which suggests that enhanced communication between providers and patients might allow for clearer explanations of risks and benefits to facilitate patient decision making that prioritizes their values.5

---

## [Decision Letter · Decision Letter 1]

4 Jul 2023

PONE-D-23-01032R1Patient Education and Extracorporeal Membrane Oxygenation Preferences of Patients and Providers in COVID CarePLOS ONE

Dear Dr. Maciejewski,

Thank you for submitting your manuscript to PLOS ONE. After careful consideration, we feel that it has merit but does not fully meet PLOS ONE’s publication criteria as it currently stands. Therefore, we invite you to submit a revised version of the manuscript that addresses the points raised during the review process.

We look forward to receiving your revised manuscript.

Kind regards,

Omar A. Almohammed, Ph.D.

Academic Editor

PLOS ONE

Reviewers' comments:

Reviewer's Responses to Questions

**Comments to the Author**

1. If the authors have adequately addressed your comments raised in a previous round of review and you feel that this manuscript is now acceptable for publication, you may indicate that here to bypass the “Comments to the Author” section, enter your conflict of interest statement in the “Confidential to Editor” section, and submit your "Accept" recommendation.

Reviewer #2: All comments have been addressed

Reviewer #3: (No Response)

2. Is the manuscript technically sound, and do the data support the conclusions?

Reviewer #2: Yes

Reviewer #3: Partly

3. Has the statistical analysis been performed appropriately and rigorously? 

Reviewer #2: Yes

Reviewer #3: Yes

4. Have the authors made all data underlying the findings in their manuscript fully available?

Reviewer #2: Yes

Reviewer #3: Yes

5. Is the manuscript presented in an intelligible fashion and written in standard English?

Reviewer #2: Yes

Reviewer #3: No

6. Review Comments to the Author

Reviewer #2: None

Reviewer #3: Ethan D. Borre and the coauthors present a interesting work, which could have value to readers of PLOS ONE and colleagues worldwide if it was again comprehensively revised.

While the introduction and material and methods are adequately presented, this is unfortunately not at all the case for the results and discussion.

Far too few results are presented in a much too short form. If the authors want to publish the manuscript in such a prestigious journal, it must be urgently revised.

The discussion is unfortunately very disappointing at the moment. It is much too superficial. Important current literature is not discussed. From my point of view, the discussion needs to be rewritten in large parts.

Likewise, the figures need a complete revision. They are currently not suitable for publication in this form, neither in terms of content nor optically.

In summary, the publication deals with a very interesting topic. If the authors want to publish this in PLOS ONE, a comprehensive revision is urgently necessary. I am already looking forward to reading this revision.

---

## [Author Response · Author response to Decision Letter 1]

10 Oct 2023

Response to Reviewer Comments on PONE-D-23-01032R1

Reviewer #2: None

Reviewer #3: 

1) Ethan D. Borre and the coauthors present a interesting work, which could have value to readers of PLOS ONE and colleagues worldwide if it was again comprehensively revised. While the introduction and material and methods are adequately presented, this is unfortunately not at all the case for the results and discussion.

Response: We appreciate the positive comments about this manuscript and have revised the manuscript as suggested by comments below.

2) Far too few results are presented in a much too short form. If the authors want to publish the manuscript in such a prestigious journal, it must be urgently revised.

Response: The reviewer’s point is very well taken in that we tersely summarize results across several patient scenarios. In the revised manuscript, we now discuss Table 2 results hypothetical patient by hypothetical patient to provide more detail as suggested.

3) The discussion is unfortunately very disappointing at the moment. It is much too superficial. Important current literature is not discussed. From my point of view, the discussion needs to be rewritten in large parts.

Response: The reviewer’s point is again well taken. We now summarize the results from this study in the context of systematic reviews (Haiduc et al. 2020 J Cardiac Surg; Ramanathan et al. 2021 Critical Care; Ling et al. 2022 Critical Care) and large multi-site cohort studies (Schmidt Lancet Respiratory Med 2023; Urner BMJ 2022) that show that age and comorbidity are positively associated with mortality.

4) Likewise, the figures need a complete revision. They are currently not suitable for publication in this form, neither in terms of content nor optically.

Response: After consultation with the editor, we have been advised to retain the 3 tables as presented in the original submission.

5) In summary, the publication deals with a very interesting topic. If the authors want to publish this in PLOS ONE, a comprehensive revision is urgently necessary. I am already looking forward to reading this revision.

Response: We again appreciate the encouraging reveiwer comments and hope that the revision is deemed responsive to comments above.

---

Response to Reviewer Comments on PONE-D-23-01032R1

Reviewer #2: None

Reviewer #3: 

1) Ethan D. Borre and the coauthors present a interesting work, which could have value to readers of PLOS ONE and colleagues worldwide if it was again comprehensively revised. While the introduction and material and methods are adequately presented, this is unfortunately not at all the case for the results and discussion.

Response: We appreciate the positive comments about this manuscript and have revised the manuscript as suggested by comments below.

2) Far too few results are presented in a much too short form. If the authors want to publish the manuscript in such a prestigious journal, it must be urgently revised.

Response: The reviewer’s point is very well taken in that we tersely summarize results across several patient scenarios. In the revised manuscript, we now discuss Table 2 results hypothetical patient by hypothetical patient to provide more detail as suggested.

3) The discussion is unfortunately very disappointing at the moment. It is much too superficial. Important current literature is not discussed. From my point of view, the discussion needs to be rewritten in large parts.

Response: The reviewer’s point is again well taken. We now summarize the results from this study in the context of systematic reviews (Haiduc et al. 2020 J Cardiac Surg; Ramanathan et al. 2021 Critical Care; Ling et al. 2022 Critical Care) and large multi-site cohort studies (Schmidt Lancet Respiratory Med 2023; Urner BMJ 2022) that show that age and comorbidity are positively associated with mortality.

4) Likewise, the figures need a complete revision. They are currently not suitable for publication in this form, neither in terms of content nor optically.

Response: After consultation with the editor, we have been advised to retain the 3 tables as presented in the original submission.

5) In summary, the publication deals with a very interesting topic. If the authors want to publish this in PLOS ONE, a comprehensive revision is urgently necessary. I am already looking forward to reading this revision.

Response: We again appreciate the encouraging reveiwer comments and hope that the revision is deemed responsive to comments above.

---

## [Decision Letter · Decision Letter 2]

4 Jan 2024

Patient Education and Extracorporeal Membrane Oxygenation Preferences of Patients and Providers in COVID Care

PONE-D-23-01032R2

Dear Dr. Maciejewski,

We’re pleased to inform you that your manuscript has been judged scientifically suitable for publication and will be formally accepted for publication once it meets all outstanding technical requirements.

Kind regards,

Omar A. Almohammed, Ph.D.

Academic Editor

PLOS ONE

Reviewers' comments:

Reviewer's Responses to Questions

**Comments to the Author**

1. If the authors have adequately addressed your comments raised in a previous round of review and you feel that this manuscript is now acceptable for publication, you may indicate that here to bypass the “Comments to the Author” section, enter your conflict of interest statement in the “Confidential to Editor” section, and submit your "Accept" recommendation.

Reviewer #3: All comments have been addressed

2. Is the manuscript technically sound, and do the data support the conclusions?

Reviewer #3: Yes

3. Has the statistical analysis been performed appropriately and rigorously? 

Reviewer #3: Yes

4. Have the authors made all data underlying the findings in their manuscript fully available?

Reviewer #3: Yes

5. Is the manuscript presented in an intelligible fashion and written in standard English?

Reviewer #3: Yes

6. Review Comments to the Author

Reviewer #3: By implementing the suggestions, the manuscript has improved significantly. Congratulations! This is now a perfect manuscript!

7. PLOS authors have the option to publish the peer review history of their article (what does this mean?). If published, this will include your full peer review and any attached files.

Reviewer #3: No

---

## [Editor Report · Acceptance letter]

1 Feb 2024

PONE-D-23-01032R2 

PLOS ONE

Dear Dr. Maciejewski, 

I'm pleased to inform you that your manuscript has been deemed suitable for publication in PLOS ONE. Congratulations! Your manuscript is now being handed over to our production team.

Kind regards, 

on behalf of

Dr. Omar A. Almohammed 

Academic Editor

PLOS ONE